# Portable FRET-Based Biosensor Device for On-Site Lead Detection

**DOI:** 10.3390/bios12030157

**Published:** 2022-03-02

**Authors:** Wei-Qun Lai, Yu-Fen Chang, Fang-Ning Chou, De-Ming Yang

**Affiliations:** 1Microscopy Service Laboratory, Basic Research Division, Department of Medical Research, Taipei Veterans General Hospital, Taipei 11217, Taiwan; willie@gm.ym.edu.tw (W.-Q.L.); carcatqqq@gmail.com (F.-N.C.); 2Institute of Biophotonics, School of Biomedical Science and Engineering, National Yang Ming Chiao Tung University, Taipei 11221, Taiwan; 3LumiSTAR Biotechnology, Inc., Taipei City 115, Taiwan; yu-fen.chang@lumistar.com.tw

**Keywords:** lead biosensors, FRET, portable Pb sensor, smartphone-based device, Met-lead, tap water lead, groundwater lead

## Abstract

Most methods for measuring environmental lead (Pb) content are time consuming, expensive, hazardous, and restricted to specific analytical systems. To provide a facile, safe tool to detect Pb, we created pMet-lead, a portable fluorescence resonance energy transfer (FRET)-based Pb-biosensor. The pMet-lead device comprises a 3D-printed frame housing a 405-nm laser diode—an excitation source for fluorescence emission images (YFP and CFP)—accompanied by optical filters, a customized sample holder with a Met-lead 1.44 M1 (the most recent version)-embedded biochip, and an optical lens aligned for smartphone compatibility. Measuring the emission ratios (Y/C) of the FRET components enabled Pb detection with a dynamic range of nearly 2 (1.96), a pMet-lead/Pb dissociation constant (K_d_) 45.62 nM, and a limit of detection 24 nM (0.474 μg/dL, 4.74 ppb). To mitigate earlier problems with a lack of selectivity for Pb vs. zinc, we preincubated samples with tricine, a low-affinity zinc chelator. We validated the pMet-lead measurements of the characterized laboratory samples and unknown samples from six regions in Taiwan by inductively coupled plasma mass spectrometry (ICP-MS). Notably, two unknown samples had Y/C ratios significantly higher than that of the control (3.48 ± 0.08 and 3.74 ± 0.12 vs. 2.79 ± 0.02), along with Pb concentrations (10.6 ppb and 15.24 ppb) above the WHO-permitted level of 10 ppb in tap water, while the remaining four unknowns showed no detectable Pb upon ICP-MS. These results demonstrate that pMet-lead provides a rapid, sensitive means for on-site Pb detection in water from the environment and in living/drinking supply systems to prevent potential Pb poisoning.

## 1. Introduction

More than 800 million children in the world live in toxic environments containing the heavy metal lead (Pb) [1]. Although the measurement of blood lead level (BLL) is the only officially approved way to evaluate the extent of Pb exposure within a person’s body, long-term studies on Pb poisoning raise questions about the so-called safe levels in children; a BLL as low as 5 μg/dL in the young population is still dangerous if it is chronic [2,3,4]. Toxic Pb may be continuously supplied through leaded pipes or other unknown sources; drinking water contaminated with Pb, even at concentrations as low as 10 ppb, can cause health problems that, sadly, go undetected in many parts of the world [5]. To avoid the silent and irreversible damage of Pb in the general population, low-cost and highly accurate Pb-detection devices for both environmental and biological (blood or urine) samples are urgently needed. Such devices will increase awareness of the threat of Pb poisoning through daily activities involving drinking, eating, and contact.

We have spent years developing and optimizing workable Pb-biosensors, i.e., genetically encoded fluorescent protein biosensors [6,7,8,9]. The sensitivity (limit of detection, LOD) of the best Pb-biosensor, Met-lead 1.44 M1, can be as low as 10 nM (around 0.2 μg/dL or 2 ppb) [8], which is sufficient for use with real samples, e.g., to determine BLL from subjects with suspected lead exposure and to assess tap water for conformity to the World Health Organization (WHO) limit for Pb (10 ppb) [10]. Thus, the application of this optimized Pb-biosensor for general use would benefit the world’s population.

The development of many portable sensing devices has progressed hand in hand with smartphone technology, especially with the advent of 4G/5G internet connectivity. The in-built complementary metal-oxide-semiconductor (CMOS) camera (with 2048 × 1536 or more pixels) used as the signal/image recorder, with either simple or complex optics, within smartphones (e.g., Nokia N73, PureView 808; Sony-Ericsson U10i AinoTM; iPhone 2G, 4G, 4S, 5, etc.) can be combined with additional suitable optical components and materials. Lenses (objective or ball lens), illumination sources (lasers or light-emitting diodes, LEDs), microfluidic channels (for small-volume samples), 3D-printed frame structures, and computer programs or algorithms can be used with smartphones to capture images of interest for a variety of purposes, such as counting the number of targeted objects. For advanced biomedical applications, clear bright-field and/or fluorescence images with an appropriate magnification (from 7×, 10×, 28× to 350×), spatial resolution (from 1.2 to 20 μm), and field of view (FOV; from 150 μm^2^ to 180 mm^2^) can be acquired through smartphone-based devices at a low cost [11,12,13,14,15]. Such smartphone-based devices have been applied in a variety of biomedical and environmental settings [16], such as the detection of glucose [17], ATP [18], inorganic phosphate [19], mercury (Hg) [20], and even certain cancer cells [21], as well as viruses such as human immunodeficiency virus type 1 (HIV-1) [22] and severe acute respiratory syndrome coronavirus 2 (SARS-CoV-2) [23].

In this study, we built a portable Pb-sensing device, pMet-lead, that can promptly detect Pb in samples by acquiring ratiometric Y/C FRET signals from the optimized Pb-biosensor Met-lead 1.44 M1. In this way, the presence of Pb in environmental water sources, such as tap water or groundwater, can be revealed in real time.

## 2. Materials and Methods

### 2.1. Instrumentation of the Portable FRET-Based Pb-Biosensor Device (pMet-Lead)

The 3D design of the FRET-based portable device for Pb detection was conducted via 3D graphing using Fusion 360 (Autodesk, Inc. San Rafael, CA, USA), according to the specifications of the selected smartphones which support RAW (DNG) data above Android 7.0, e.g., Asus ZenFone 3 (ASUSTeK Computer Inc. Taiwan; with Sony IMX298, a 16 MP camera, 4608 × 3456 pixels under Android 7.0) and/or Xiaomi Mi11 Lite 5G (Samsung ISOCELL GW3, a 64 MP camera under Android 11) (Smartphone camera info: https://zh.wikipedia.org/wiki/Exmor, accessed on 22 February 2022), sample holder, and optical distance for fluorescent signal recording. The size of the device was around 190 mm × 140 mm × 105 mm (L × W × H; Figure 1 and Appendix A, see Appendix A). Through the additive manufacturing (AM) technique, two printing machines, CR-5 Pro (Shenzhen Creality 3D Technology Co, Ltd., Shenzhen, China) and CR-8 (Shenzhen Creality 3D Technology Co, Ltd., Shenzhen, China), were used. The printing material polylactide (PLA) was applied at a printing temperature range of 190–215 °C, under the control of Cura software (Ultimaker B.V., Utrecht, The Netherlands). The overall procedure for manufacturing this device required approximately 50 h.

### 2.2. Preparing of the Met-Lead-Containing Biochip

Single human living cells from different sources were used to examine the sensing ability of Met-lead biosensors. Human cell lines—human embryonic kidney cells (HEK293) and human cancer cells (HeLa)—were obtained from the American Type Culture Collection (ATCC, Rockville, MD). Cell lines were cultured according to the culture manual [6,24], and prepared for final observation with Tyrode’s buffer. For further FRET ratio imaging, all human cells were seeded on 24 mm coverglasses (Deckglaser) coated with poly-L-lysine and were transfected with the gene encoding Met-lead using lipofectamine (Invitrogen, Thermo Fisher Scientific Inc., Waltham, MA, USA) according to the manual [6,8,25,26]. The cells were used for lead biosensing 2 days after transfection.

### 2.3. Sample Source and Preparation

The tested samples were obtained from water supply systems, such as tap water or groundwater for drinking and other living usage. Tap water and groundwater were collected from an inhabited area near Taipei (Taoyuan County), and from a rural area of Taiwan (Dongshi Township, Yunlin County). For standard samples, different concentrations of Pb solutions (1, 10, 20, 30, 40, 50, and 100 nM, and 1 and 10 μM) were prepared as previously described [8]. Sample were prepared for validation (Section 2.6) as described previously [8].

### 2.4. Ratio Imaging under Epifluorescence FRET Microscope

Met-lead-expressing cells were placed on an inverted live-cell FRET microscope [7,8,10]. To detect the epifluorescence signals of cp173Venus (yellow fluorescent protein; YFP) and ECFP(ΔC11) (cyan fluorescent protein; CFP) from Met-lead, the FRET emission (Y/C) ratio image system was equipped with an inverted microscope (Axiovert 200M; with 10× objectives, NA = 0.3, Zeiss, Germany), a W-View module (Gemini, Hamamatsu, Japan; with emission filters 542/27 nm for YFP and 483/32 nm for CFP), and a CMOS camera (ORCA-Flash 4.0 LT, Hamamatsu, Japan) controlled by the HCImage software. The YFP (530–630 nm) and CFP (460–500 nm) emission signals were acquired continuously through the W-View module [7,8,10].

### 2.5. Ratio Imaging under pMet-Lead

For portable FRET ratio imaging, biochips containing Met-lead were placed into the sample holder of pMet-lead as described in Section 2.1, with a 405 nm laser used as the excitation source. To make it possible to obtain the emission signals of cp173Venus and ECFP(ΔC11) from a biochip containing Met-lead, the device contained switchable filters, 542/27 nm and 483/32 nm, for YFP and CFP, respectively. The fluorescent signals were recorded using the built-in camera in the smartphone (ZenPhone 3, Asus, Taiwan; or Xiaomi Mi11 Lite 5G, Mi, China), which was controlled by an app capable of collecting raw data (10-bit images). The emission signals of YFP (530–630 nm) and CFP (460–500 nm) were acquired separately. Before the Y/C ratio is calculated and data were displayed from the iMet-lead program (Figure 2) in a MATLAB environment (Figure 2A,B), and/or on a smartphone (Figure 2C and Appendix A), the images taken from the smartphone were pre-processed through an Algorithm 1:
**Algorithm 1** Pre-processing of RAW (DNG) images taken with a smartphone cameraIntensity or value of YFP, CFP, and Ratio are Y(i,j), C(i,j), and R(i,j) respectively.The pixel position in length and width is i,j.for each pixels intensity (i = 2:length, j = 2:width)if Y(i,j) below 200 then Y(i,j)= 1/4(*Y*(*i* − 1,*j*) + *Y*(*i* + 1,*j*) + *Y*(*i*,*j* − 1) + *Y*(*i*,*j* + 1))if C(i,j) below 150 then Y(i,j)= 1/4(C(*i* − 1,*j*) + C(*i* + 1,*j*) + C(*i*,*j* − 1) + C(*i*,*j* + 1))if R(i,j) below 1 then Y(i,j)= 1/4(R(*i* − 1,*j*) + R(*i* + 1,*j*) + R(*i*,*j* − 1) + R(*i*,*j* + 1))end

### 2.6. Validation of the Y/C Ratio Data from pMet-Lead

Samples from both laboratory and real environments, such as tap water, were tested by the general standard method, i.e., inductively coupled plasma mass spectrometry (ICP-MS); the tests were performed within the Division of Clinical Toxicology and Occupational Medicine, Taipei Veterans General Hospital.

### 2.7. Data Analysis

ImageJ was used to combine the ratio of fluorescent signals of cp173Venus and ECFP(ΔC11) (ratio plus), and displayed with a rainbow color plate to visualize the FRET biosensing from blue to red (look-up table, LUT) within tested biochips. All experiments were carried out three times with at least three different sets of tested samples (tap water or groundwater). Data gathered from the different batches of Met-lead biochips on different samples were integrated to calculate the emission (Y/C) ratios. Significant changes were indicated based on the *p*-value calculated using a two-tailed *t*-test function contained in the IBM SPSS statistics. The mean differences are described as significant at the 0.0005 (***) or 0.005 (**) levels.

## 3. Results

### 3.1. Design and Outline of the Portable FRET-Based Pb-Biosensor Device (pMet-Lead)

Building on previous successes with FRET-based Pb biosensing using Met-leads [7,8,10], we designed and manufactured a portable device (the pMet-lead) that is compatible with smartphones and has the capacity to acquire Y/C emission ratio images from Met-leads or other FRET biosensors (Figure 1 and Appendix A in detail). Our aim in building this portable FRET device was to allow FRET biosensing to be performed without the need for special FRET microscopes, which are only available in research or hospital laboratories. This pMet-lead device comprises two parts: (1) A Pb-sensing device (Figure 1A,B) containing the following: a pair of illumination sources, an LED (for bright-field images; BF in Figure 1C) and a 405 nm laser diode (for donor excitation during the FRET ratio biosensing); a pair of convex lenses; a sample holder with a biochip for Pb biosensing; and emission filters (for obtaining the Y/C ratio images, Figure 1C). We performed functional tests on the biochips containing the Met-lead biosensors in regard to both imaging and sensing ability under a well-defined FRET microscope (Appendix A); (2) A holder to contain an Android smartphone linked to iMet-lead, driven either by notebooks through a USB connection in a MATLAB environment (Figure 2A,B) or by smartphones with an appropriate app preinstalled (Figure 2C and Appendix A). To complete the setup of this smartphone-based Pb-sensing device, we seeded the cells expressing Met-lead 1.44 M1 onto the biochip and inserted it into the device (Figure 2C).

As pMet-lead can obtain both BF and fluorescence (YFP and CFP) images of the Pb-biosensor—the Met-lead-coated biochip—with a smartphone camera (Figure 1), the next step toward portable Pb detection was to enable the device to register the presence of Pb through the analysis of the YFP/CFP ratios from the Pb-biosensor, as previously demonstrated [7,8,10]. To detect Pb in the samples, we developed a software package (for notebooks) and an app (for smartphones), both called iMet-lead, that perform the Y/C ratio transformation step described in the flowchart in Figure 2A (Appendix A for smartphone); this step can be controlled using the options provided on the iMet-lead interface. The visualized ratios (displayed in rainbow color; Figure 2B) derived from the YFP and CFP images obtained with the smartphone camera can be easily interpreted by the surveyor in real time through the iMet-lead interface (Figure 2B and Appendix A). Figure 2C demonstrates the full procedure for using pMet-lead for fast detection of Pb in environmental water or water supply systems, such as tap water (for drinking) and groundwater. Overall, pMet-lead can quickly process the images representing Pb levels and report the presence of Pb on the notebook or smartphone screen.

### 3.2. Performance of pMet-Lead

Built-in smartphone cameras have a depth of 8 bits (2^0^–2^8^, which is a relatively small dynamic range) in each RGB channel for displaying color images (Appendix A). In contrast, scientific-grade cameras produce images with at least 12 bits (2^0^–2^12^, providing a relatively large dynamic range) in monochrome format. Some smartphones produce images with a depth of more than 10 bits, but only in the raw data format (DNG). Thus, we acquired the fluorescent images of samples from the cameras of smartphones by extracting the DNG-format files (Appendix A). However, upon close inspection of the fluorescent images taken from pMet-lead processing, repeated black spots could be found in both YFP and CFP channels (Figure 3A). These spots in the DNG file format seemed not to derive from the device itself (Appendix A) and could generate large errors in the Y/C ratios, since they are not true fluorescence signals from the Met-lead biosensor; they may also be the cause of the poor image quality of Y/C ratio color images (Figure 3A). Therefore, as an alternative to using DNG files, we figured out a method/function, as described in Section 2.5, to recalculate the signals from the YFP and CFP channels together (Figure 3B); this was similarly used to correct the Y/C ratio taken from pMet-lead (Figure 3C). We found that this preprocessing improved the quality of both the fluorescent and Y/C ratio color images sufficiently to enable further characterization of the sensing ability of pMet-lead (Figure 3D).

We put the biochip containing the Met-lead 1.44 M1-expressing HEK293 cells into the device and proceeded with Pb sensing (Figure 4). Similar to previous observations (Figure 1C), including the pre-processing step led to better fluorescent images from the YFP and CFP channels of the Met-lead biosensor, as shown on the screen of the notebook/smartphone (Figure 4A). The Y/C ratio color images of the same captured field was also promptly generated (Y/C ratio of Figure 4A). The emission ratios within the Met-lead-expressing samples were around 2.7969 ± 0.1001 for water without Pb (control condition, basal level). When various concentrations of Pb were presented, the ratios increased significantly from 2.9136 ± 0.800 (1 nM) and 2.8310 ± 0.0112 (10 nM) to 3.5133 ± 0.0782 (40 nM), 4.3811 ± 0.0994 (50 nM), 4.8107 ± 0.0470 (100 nM), 5.1732 ± 0.0298 (1 μM), and up to 5.5572 ± 0.0138 (10 μM) (Figure 4B). Based on these results, we determined that the dynamic range of this new Pb-sensing device was 1.96. The linearity (R^2^) of known samples for ICP-MS to pMet-lead was 0.9169 at low concentrations (from 10 nM to 100 nM) under a probability scale, and 0.9997 at high concentrations (from 100 nM, 1 μM to 10 μM) under a log scale (Figure 4C). Through titration experiments, we determined that the dissociation constant (K_d_) of the Met-lead 1.44 M1 biosensor for Pb in this device was 45.62 nM (Appendix A), similar to that reported for Met-lead 1.44 M1 on a FRET microscope with 20× objectives (25.97 nM) [8]. The LOD of Met-lead 1.44 M1 for Pb in a high-cost FRET microscope is 10 nM [8], while the LOD of the new portable device was calculated as 4.74 ppb (24 nM) (Figure 4C). The data from pMet-lead at different concentrations of Pb were further validated by ICP-MS (Figure 4C). The ratio was 2.8310 ± 0.0112 at around 2.85 ppb (10 nM), 3.5133 ± 0.0782 at 9.6 ppb (40 nM), and 4.8107 ± 0.0470 at 23.35 ppb (100 nM) (Table 1).

### 3.3. Solution to the Selectivity Issue of Met-Leads and pMet-Lead

In the first version of Met-lead we developed (Met-lead 1.59), the sensor’s selectivity for zinc (Zn) as well as Pb ions posed a problem [6], and this remained a concern with the newer version (1.44 M1) [8]. Therefore, in order to perform practical testing of the Met-lead 1.44 M1 biosensor for detecting environmental Pb, we had to overcome the Zn interference. We tested whether the use of tricine, a low-affinity Zn chelator [27,28] that does not bind Pb, could mitigate this problem. We found that a test of pure water containing Zn (11 μM, or around 1500 μg/dL; three times higher than the U.S. EPA regulation of 500 μg/dL, or 5 ppm, or 3.33 μM) with Met-lead 1.44 M1 led to an increase in the Y/C ratio from a resting value of 1.6008 ± 0.0451 to 2.4474 ± 0.1034 (solid line in Figure 5A; full time-scale in Appendix A). An 11 μM solution of Zn in water, that was pre-incubated with tricine (10 mM), yielded a significantly lower ratio increase, from 1.6225 ± 0.0374, but it remained around 1.7400 ± 0.0375, indicating that the increase in ratio was Zn-induced (dash line in Figure 5A; full time-scale in Appendix A). The pre-incubation of tricine (10 mM) and Pb (10 μM) with the same Zn-containing water sample elevated the ratio from 1.7388 ± 0.060 up to 3.1545 ± 0.1102 (dotted line in Figure 5A; full time-scale in Appendix A), demonstrating that the sensor could still sense Pb. The averaged data summarizing the tricine tests indicated that tricine could be a good adjuvant for Pb detection through the Met-lead Pb-biosensor (Figure 5B).

Next, we took a tap water sample from a faucet in our laboratory as the first practical test of the sensor (Figure 5C). Using ICP-MS, this water sample was confirmed to contain a small amount of Zn (97.8 μg/dL; lower than the U.S. EPA regulation 500 μg/dL) and Pb (9.75 nM, or 0.195 μg/dL; lower than the WHO regulation 1 μg/dL) (Appendix A). This water sample produced a significant increase in the Y/C ratio through our Met-lead biosensor, from the resting value of 2.2903 ± 0.0609 to 3.8879 ± 0.0861 (solid line in Figure 5C). When the same water sample was pre-mixed with tricine (10 mM), this increased the Y/C ratio of the Met-lead biosensor from 2.0512 ± 0.0970 to 2.6407 ± 0.0604 within 10 min (dash line in Figure 5C; full time-scale in Appendix A). These data summarize the results of the tests with real tap water samples (Figure 5D) and suggest that Met-leads and pMet-lead can be successfully used in real-world applications. The animated data on these tricine tests are shown in Appendix A.

### 3.4. Practical Water Check by pMet-Lead

Next, we further tested the practical application of the newly developed pMet-lead (Figure 6; Table 2). Water samples gathered from six selected regions of Taiwan (Figure 6A), including inhabited areas near Taipei city (sample I and II), Taoyuan County (sample III), Miaoli County (sample IV), Nantou County (sample V), and even the rural area of Taiwan, i.e., Dongshi Township, Yunlin County (sample VI). These water samples were placed into the pMet-lead to obtain Y/C emission ratio color images (Figure 6B) and ratios (Figure 6C). Two of these samples had significantly higher Y/C emission ratios (3.4803 ± 0.0839 in sample IV; 3.7431 ± 0.1236 in sample VI, Figure 6C) than the control set (2.7931 ± 0.025, Table 2). The remaining samples had low Y/C ratios, which were confirmed by ICP-MS (Table 2) to be Pb-negative or contain relatively low Pb: No lead was detected (NA) in samples I (2.3136 ± 0.0416), III (2.4554 ± 0.05369), and V (2.3581 ± 0.0258), and 0.64 ppb Pb was detected in sample II (2.2897 ± 0.0250). The samples with high Y/C ratios were also confirmed to contain Pb exceeding the acceptable level (10 ppb) set by the WHO (10.6 ppb in sample IV; 15.24 in sample VI, Table 2). The linearity (R^2^) of ICP-MS to pMet-lead for the six selected real-world unknown samples was 0.979 (Figure 6D). Therefore, these two samples were proven to contain Pb. The data for the rest of the tested samples are shown in Table 2.

## 4. Discussion

The integration of smartphones into sensing methods has been widely applied, whether in fluorescence or colorimetric systems [14,16,17], or through sensing chips or paper formats [19,20,21], to fulfill demands due to the large number of users [12,13,18,24,29,30,31]. However, it is relatively rare to see FRET-based devices invented for the development of portable fluorescence sensing. Possible reasons for this could be: (1) two or more fluorescent signals need to be measured simultaneously or alternatively, which causes a major inconvenience for the hardware design of sensing devices for both laboratory-based and portable, smartphone-based microscopic imaging and sensing; (2) further processing procedures are required, e.g., the YFP/CFP ratios from the image data taken from the portable device require processing through the division operation; (3) the increase in the ratio representing the presence of target molecules within FRET-based biosensors is restricted to less than 10-fold, whereas typical fluorescence methods for sensing target binding can detect fluorescence increases of 1000-folds.

Despite the above disadvantages, FRET-based biosensors have some benefits for monitoring specific targets [32,33,34,35,36]. First, the ratios represent the meanness of normalization. This property of FRET ratio detection provides superior data for indicating the existence of a target, since there is no need to find a standard baseline, as is required for most indicators. Briefly, each fluorescent sensor emits a specific emission signal, and the increase (or decrease when it is an off sensor) in the changes in fluorescence intensity needs to be normalized in order to pool these data, since the original intensity of each molecule and the corresponding increase in fluorescence intensity for each one are both different. This might be why Met-lead 1.44 M1 is sensitive enough to detect Pb contents as low as 10 nM [8]. Secondly, the rainbow color plate used for the presentation of FRET-based sensing is an easy way to visualize the detection results: red means polluted by Pb, while blue means OK to drink if the tested sample is drinking water (Figure 6B). Having a portable Pb-sensing device on hand provides the immense benefit of not having to go through the costly and time-consuming process of collecting and transporting water samples to testing companies.

Long-term chronic exposure to low Pb increases the risk of cardiological and neurological diseases in humans [37,38,39,40,41,42]. Currently, the only validated way to determine the Pb content of suspicious targets is using standard precision analytical instruments, such as ICP-MS, which are available only in hospitals or testing companies, which are costly and require at least two weeks before results are known [2,3,43]. Thus, any kind of portable device that can monitor Pb in drinking water supplies, such as tap water or groundwater, would be helpful in preventing Pb poisoning in a community. Here, we developed a portable Pb-sensing device that is compatible with smartphones and capable of distinguishing Pb from Zn (Figure 4) to improve the practicality of Met-leads. Furthermore, pMet-lead has a relatively high performance/cost ratio (Table 3) for on-site, real-time quantitative environmental detection of Pb (LOD: 9.6 ppb) (Figure 2).

On the issue of metal selectivity, copper (Cu), mercury (Hg), and cadmium (Cd) might interfere with the Pb detection of Met-lead in addition to Zn. It would be necessary to have a proper solution in the case of applying pMet-lead to wastewater Pb detections. The strategy of using metal chelators similar to tricine (Zn), such as penicillamine to Cu [51], DMPS to Hg [52], and 3-((5-(trifluoromethyl)-1,3,4-thiadiazol-2-yl)amino)benzo[d]isothiazole 1,1-dioxide (AL14) [53] to Cd could be a good solution to integrate into pMet-lead in the future.

Finally, the biochip containing the Met-lead 1.44 M1-expressing living cells is the best one that we can reliably provide at present, but could be optimized further. To further improve portability, the biochip could be coated with purified Met-lead proteins, subsequently lyophilized for longer storage, e.g., for one year [17]. The solubility and multimeric nature of Met-lead might lower the Pb-sensing ability during purification and resuspension (after lyophilization) into aqueous solution [6,8]. An alternative to this biochip is perhaps the single fluorescent protein version of the Met-lead biosensor, which will be a good choice in the future for better purification and storage, as in the previous successful case of the single fluorescent protein GCaMP Ca biosensor [54]. Another possibility is to use living bacteria expressing Met-lead on their surface [55,56]. Such a bacterial version of the smartphone-based Pb device may enable the quick determination of Pb pollution anytime, anywhere. The limitations of using prokaryotes, such as Escherichia coli, are similar to those with human cell lines, i.e., difficulties in storing and maintaining the biochip, but bacteria can be stored in 4 °C refrigerators and can have relatively prolonged metabolic activity compared to eukaryotic cells.

## 5. Conclusions

In this study, we developed a portable biosensor device, pMet-lead, for on-site Pb detection. Smartphones are a useful tool capable of running the iMet-lead app to take photos of samples; geographical information for the tested samples can also be provided along with the Pb ratio to improve data integrity, i.e., automatic collection of geographic information through the cellphone’s GPS-based geographic systems. These benefits increase the practicality of pMet-lead, by not only helping governing bodies identify and thus eliminate the illegal leakage of heavy metal lead into environmental water, but also by encouraging citizens to be made aware of the status of their daily drinking water without having to spend a lot of money and wait more than two weeks for the results. In summary, this study demonstrates the use of pMet-lead, a smartphone-compatible device for the biosensing of Pb, as an on-site tool to detect environmental lead exposure.

## Figures and Tables

**Figure 1 biosensors-12-00157-f001:**
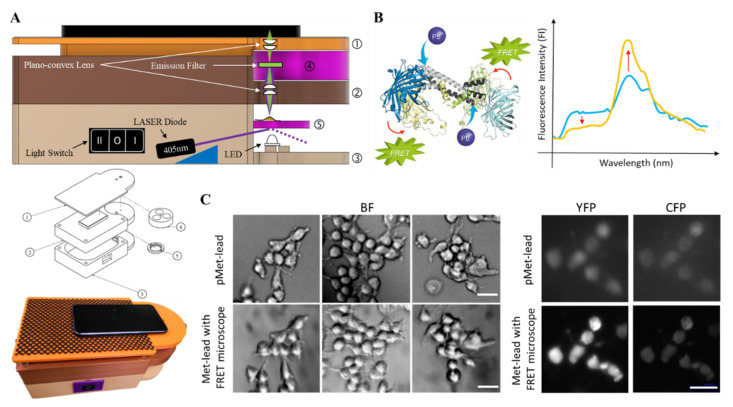
Design and layout of the portable FRET-based Pb-sensing device (pMet-lead), and examples of images generated with pMet-lead compared to conventional FRET ratio imaging. (**A**) Side view (upper), exploded view (middle), and entity (bottom) of the optical design of the pMet-lead device to demonstrate the general setup of pMet-lead: (1) the top plate holds the smartphone; (2) the middle part of the apparatus contains the plano-convex lens and is linked to the filter turret (part 4); (3) the bottom part contains the illumination sources (LED for BF, LASER Diode for FL and light switch to turn off (O) or on to BF (I) or FL (II)), and is linked to the sample stage (part 5); (4) the filter turret contains two emission fluorescence filters that receive YFP and CFP emission signals (changeable according to various FRET pairs); (5) the sample stage receives the biochip containing the cells expressing the Met-lead biosensor. (**B**) Molecular structure design (left, originally from [8], and permission has been obtained to use the same here) and emission spectrum (right) of FRET-based Met-lead. (**C**) The representative bright-field (BF, left) and fluorescence (YFP, middle; CFP, right) images of the Pb-biosensor-expressing HEK293 cells obtained with pMet-lead (above) or under a FRET microscope (with 10 × objective; below). Scale bar, 30 μm.

**Figure 2 biosensors-12-00157-f002:**
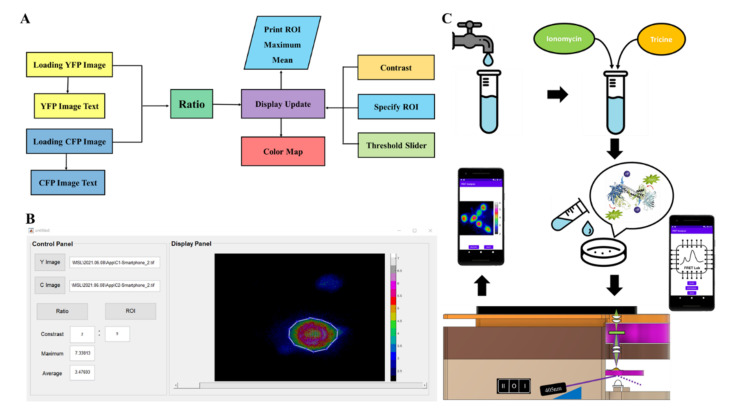
Processing of fluorescence images in pMet-lead. (**A**) Flowchart of the computational window graphical user interface (GUI) for ratio color imaging. After image input (loading) of YFP and CFP, select ‘ratio’ to make and save the ratio images to temporary memory sites. The calculated ratios will be displayed in a rainbow color format (display update; color map; maximum mean). The ranges of selected regions of interest (ROIs) can be manually adjusted to show the best present status (specify ROI; contrast; threshold slider). (**B**) Screen shot of the GUI for the FRET-based device interface during Pb sensing. The ratio color bar represents YFP/CFP ratios from 2 (black) to 7 (white). (**C**) The whole procedure for on-site Pb sensing with pMet-lead.

**Figure 3 biosensors-12-00157-f003:**
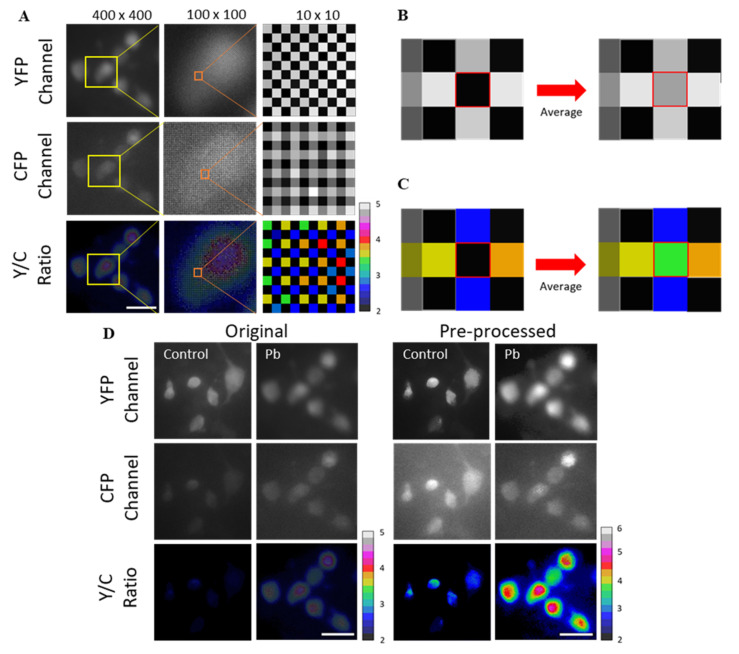
Pre-processing of pMet-lead fluorescence images taken with a smartphone camera. (**A**) Met-lead-expressing cells taken from pMet-lead were first separated into YFP and CFP channels. The Y/C ratio image was obtained through the division of fluorescence from the YFP and CFP channels. Selected regions of YFP, CFP, and Y/C ratio from an image size of 400 × 400 pixels (left) were further magnified to 100 × 100 pixels (middle) and 10 × 10 pixels (right). Black spots can be observed from both the YFP and CFP channels. (**B**,**C**) A conceptual illustration of the function of the image processing step described in Section 2.5 to fill black spots with signals, from adjacent regions of fluorescent YFP or CFP signals (**B**), or Y/C ratios (**C**). (**D**) Comparison of images without (control) or with Pb (100 nM) taken from pMet-lead without (original) and with (pre-processed) corrections as described in Section 2.5. Scale bars, 30 μm. The ratio color bars are from 2 to 5 for the original images (left), and from 2 to 6 for the pre-processed images (right).

**Figure 4 biosensors-12-00157-f004:**
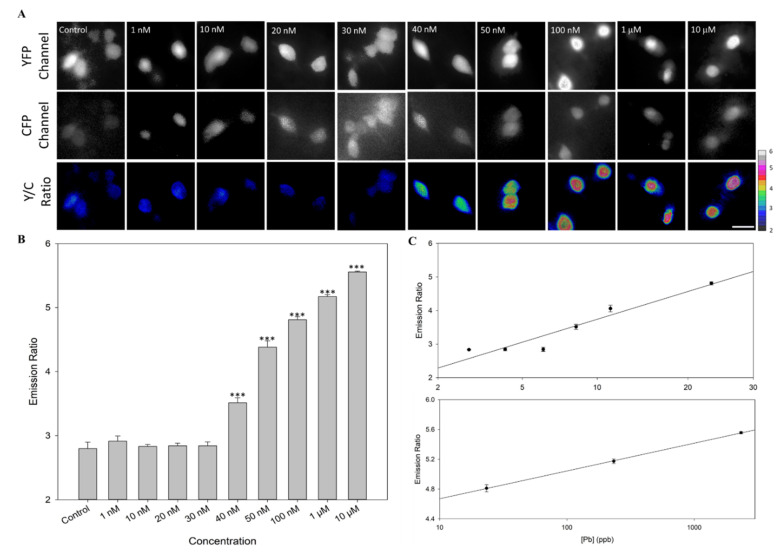
Sensing ability of pMet-lead. (**A**) The representative pMet-lead FRET ratio images of HEK293 cells expressing Met-lead 1.44 M1 taken at various concentrations of Pb from nM (1, 10, 20, 30, 40, 50, and 100 nM) to μM ranges (1 and 10 μM). The fluorescence images of Met-lead 1.44 M1 in the YFP and CFP channels, and the ratio color images (through the preprocessing procedure described in Figure 3) under CFP-specific illumination (405 nm). (**B**) Bar graphs of FRET ratios at various concentrations of Pb using pMet-lead analysis. The mean differences are described as significant at the 0.0005 (***) level. (**C**) Validation of pMet-lead (Y/C emission ratio) using ICP-MS (Pb measured in ppb) at various concentrations of Pb (upper: 10, 20, 30, 40, 50, and 100 nM; lower: 100 nM, 1 μM, and 10 μM). Scale bar, 30 μm. The ratio color bar is from 2 to 6.

**Figure 5 biosensors-12-00157-f005:**
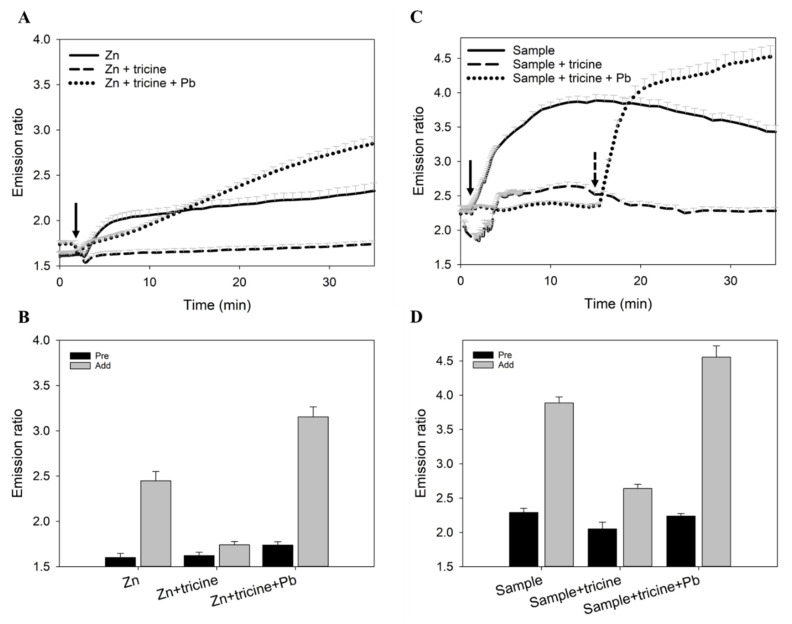
Removal of zinc with tricine enables practical Pb sensing by pMet-lead. (**A**) Time-lapse of the emission ratio (YFP/CFP) recorded by the Met-lead biosensing system. Double-distilled water (distillation-distillation H_2_O, ddH_2_O) from the laboratory was pre-mixed with 11 μM zinc alone (Zn; solid line), 11 μM zinc and 10 mM tricine (Zn + tricine; dashed line), or 11 μM zinc, 10 mM tricine, and 10 μM Pb (Zn + tricine + Pb; dotted line), and introduced at the time point indicated by the arrow. (**B**) Bar graphs of the averaged Y/C ratio values from Met-lead under the conditions in (**A**). (**C**) Samples of tap water from a laboratory faucet, without (sample; solid line) or with 10 mM of tricine added (sample + tricine; dashed line), and 10 μM Pb (sample + tricine + Pb; dotted line) were placed in the Met-lead biosensing system at the time point indicated by the arrow. All experimental sample tests were with ionomycin (5 μM). (**D**) Bar graphs of the averaged Y/C ratio values from Met-lead under the conditions in (**C**).

**Figure 6 biosensors-12-00157-f006:**
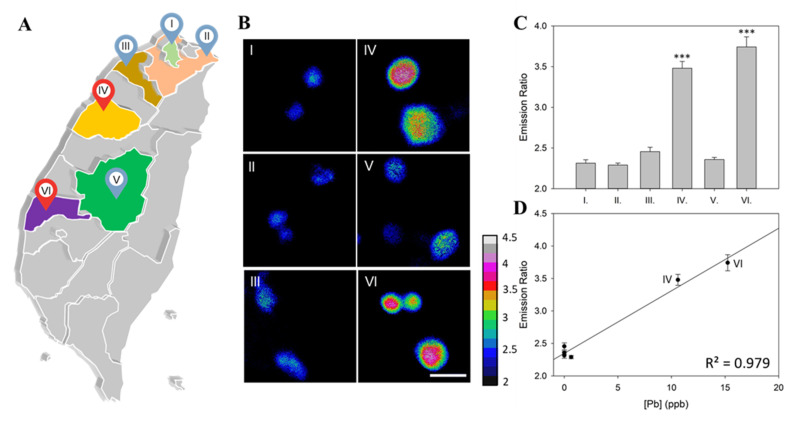
Measurement of Pb levels in selected environmental areas using the portable FRET-based Pb-sensing device. (**A**) Geographical regions where water samples were randomly collected in selected cities of Taiwan: I. Taipei city; II. New Taipei; III. Taoyuan; IV. Miaoli; V. Nantou; VI. Yunlin. (**B**) Representative ratio color images of samples I–VI analyzed using pMet-lead. (**C**) Average ratio for each geographical region compared to control water samples. The mean differences are described as significant at the 0.0005 (***) level. (**D**) pMet-lead FRET ratio values of Pb measurements from various geographic sources of water were validated by the general standard method (ICP-MS). Scale bar, 30 μm. The ratio color bar is from 2 to 4.5.

**Table 1 biosensors-12-00157-t001:** Validation of pMet-lead on Pb measurements. The FRET ratios obtained from pMet-lead Pb measurements from various sources of water (Figure 4) were validated by the general standard method (ICP-MS).

Water Samples	Mean of Y/C Ratio (with Standard Error)	ICP-MS (ppb)
Control	2.7969 ± 0.1001	NA ^1^
1 nM	2.9136 ± 0.0800	NA
10 nM	2.8310 ± 0.0112	2.85
20 nM	2.8428 ± 0.0411	4.20
30 nM	2.8413 ± 0.0611	6.14
40 nM	3.5133 ± 0.0782 (***) ^2^	8.32
50 nM	4.3811 ± 0.0994 (***)	11.18
100 nM	4.8107 ± 0.0470 (***)	23.35
1 μM	5.1732 ± 0.0298 (***)	231.45
10 μM	5.5572 ± 0.0138 (***)	2327.71

^1^ NA: none detected; ^2^ The values are below the WHO 2017 cutoff for Pb in tap water of 1 μg/dL (10 ppb, 50 nM); ***, *p* < 0.0005, two-tailed *t*-test.

**Table 2 biosensors-12-00157-t002:** Pb detection in different water supply systems. pMet-lead FRET ratios of practical Pb measurements of various sources of water in this study (Figure 6) were validated using the general standard method (ICP-MS).

Water Samples	Mean of Y/C Ratio (with Standard Error)	ICP-MS (ppb)
Control	2.7931 ± 0.025	NA ^1^
I	2.3136 ± 0.0416	NA
II	2.2897 ± 0.0250	0.64
III	2.4554 ± 0.05369	NA
IV	3.4803 ± 0.0839 (***) ^2^	10.6
V	2.3581 ± 0.0258	NA
VI	3.7431 ± 0.1236 (***) ^2^	15.24

^1^ NA: none detected; ^2^ The values are above the WHO 2017 cutoff for Pb in tap water of 1 μg/dL (10 ppb, 50 nM); ***, *p* < 0.0005, two-tailed *t*-test.

**Table 3 biosensors-12-00157-t003:** Pb detection in different water supply systems. pMet-lead FRET ratios of practical Pb measurements from various sources of water in this study (Figure 6) were validated using the general standard method (ICP-MS).

Methods	LOD	Report Time	Performance/Cost Ratio	Note
ICP-MS	0.003–0.01 ppb ^1^	Long (lab/hospital) ^1^	Low (high price)	Gold standard
GFAAS	1–10 ppb ^2^	Long (lab/hospital) ^2^	Low (high price)	Gold standard
PB200	~0.1 ppb ^3^	Real time (on-site)	High; no expandability	Not validated
PC2700	200 ppb	Fast (lab)	Low (poor LOD)	Not validated
Half-salamo-based chemosensor, HL [44]	~11 ppb (56.5 nM), cal.	Fast (lab)	Medium (spectrophotometer); expandability	Not validated
Organic immobilized composite sensor [45]	0.24 ppb (1.2 nM)	N. A. (lab)	Medium (color); no expandability; remove; resuses	Validated (ICP-AES)
Graphene oxide (GO) nanocomposite [46]	0.28 ppb (1.41 nM)	N. A. (lab/on-site)	High (voltametry); expandability	Validated (AAS)
Alk-Ti_3_C_2_ MXenes [47]	~8.2 ppb (41 nM)	Fast (lab)	Medium (SWASV); expandability	Not validated
CuZrO_3_ nanocopmposites [48]	0.1 ppb	Fast (lab)	Medium (SWASV); expandability	Not validated
SERS sensor AB18C6-gold nanoparticles [49]	0.69 pM	Fast (lab)	Medium (SERS); resuses	Validated (ICP-MS)
NiO/rGO Nanocomposite [50]	2 ppb (10 nM)	Fast (lab/on-site)	High (SWASV); expandability	Validated (ICP-AES)
Met-lead 1.59 [6]	100 ppb (500 nM)	Fast (lab)	Low (poor LOD)	Not validated
Met-lead 1.44 M1 [8]	2 ppb (10 nM)	Fast (lab)	Medium (FRET fluorescence microscope); expandability	Validated (ICP-MS)
pMet-lead (this study)	4.74 ppb (24 nM)	Real time (on-site)	High; expandability	Validated (ICP-MS)

^1^ NIEA W313.53B, Taiwan; SGS; 3 ppb for water, 10 ppb for foods; 2 weeks for report; ^2^ NIEA W303.51A, Taiwan; 2 weeks for report; ^3^
http://www.cleaninst.tw/PB200.htm (accessed on 22 February 2022); SWASV: square wave anodic stripping voltammetry; SERS: surface-enhanced Raman spectroscopy.

## Data Availability

Not applicable.

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
