# Peer review of "Portable FRET-Based Biosensor Device for On-Site Lead Detection"

_biosensors, 2022, doi:10.3390/bios12030157_

Round 1

Reviewer 1 Report

This paper talked about the design and development of a novel portable fluorescence resonance energy transfer (FRET)-based lead biosensor. Overall, this paper presented high quality fluorescence images and well-planned analytical experiments on lead sensing, interference study and six on-site sample quantification. All the content and figures are easy to understand. Optical method-based lead biosensor demonstrated in this work has good novelty and comparable sensitivity when comparing with traditional methods.

 Line 276 H2O should be H2O

Author Response

We thank Reviewer’s comments. We have corrected this typo.

Reviewer 2 Report

In the paper titled, ' Portable FRET-based biosensor device for on-site lead detections.' Lai et. al has presented a new portable devise that can be used for onsite lead detection. While this is a nice study, it lacks in originality and novelty. The core idea behind the FRET sensor had been developed by the group and had been used in multiple papers from the group before. The newer addition here is portability of the devise. While that is important for onsite studies, it is not novel. Moreover, the study has quite a few major shortcomings that need to be carefully revised before I can recommend it for publication. Here are some of my criticisms.

  1. The use of a smartphone camera for detection is used almost as a gimmick here. Specially given the fact that the instrument need to be designed and calibrated differently based on the smartphone specification (Page 2 line 75). So one can arguably put a fixed camera screen and use the app to analyze the data from the camera.
  2. Use of the smartphone makes the life difficult for the authors as the camera seems to have a wired checkerboard pattern on top of the fluorescence image. The averaging somewhat corrects for it but makes the images blurry.
  3. Can the authors comment on whether the pattern is consistent from image to image or it changes. How about between YFP and CFP channels of the same image? Can you make a direct comparison of YFP-only cells and CFP-only cells to make a baseline and use that as the a background correction?
  4. Fig1. Please describe fig. 1B and point to each component including the BF LED. Please add a scale bar for the Fig.1C top panel. Why does the image magnification sees different for BF and fluorescence images?
  5. Although the authors have referred to their old paper but, the central sensing method here needs to be described better and with illustration images explaining the conformational change upon lead binding.
  6. Fig.3 Why is there no checkerboard pattern on the control images and only in lead images? in Fig. 3A it seems like the pattern is seen only when we zoom in. However, Fig. 3D seems to have the pattern even in the zoom out version. Please explain.
  7. Fig. 4 Calibration experiment is poorly done. Why is there only four data points? there are at least seven data points in the bar diagram? the calibration curve is clearly non linear and the R2 value would not be close to 0.8 if all points were considered? Please show all data and comment on the linear range of detection, limit of detection, and dynamic range.
  8. Zn based experiments does not specifically conclude how these detection parameters are affected by the presence of Zn in water. Please provide a direct comparison (in the lab setting) if you have 0, mid and high level of Zn, how lead detection is affected? 

Author Response

Response: The original purpose of developing FRET-based Met-leads was to understand the details of Pb toxicology within living organisms under various kinds of advanced microscopes (for research only). The improvements of Met-leads allowing it to sense Pb at very low concentrations (i.e. 10 nM; 2 ppb in LOD; examined within a research laboratory) gave us great confidence to ask if it can be further applied to the environmental Pb detection. Therefore, the following goal of our study was to let this optimized Pb sensor be practically applied for Pb prevention in the real world, despite the novelty of using Me-lead 1.44 M1 might be a little bit low. On the other hand, to make this kind of FRET-based biosensors workable out of the laboratory without the research grade fluorescence microscope is another huge challenge, since it is relatively rare in the development of FRET-based portable devices. In this study, we achieved a milestone of progress for the above mentioned aims through combining the optimized biosensors into an affordable smartphone-compatible device so that even drinking water for daily life can be tested whether containing excess Pb, as Figure 2C indicates. We thank the reviewer appreciating the importance of this study.
Response 1: We thank the reviewer's comments. We added the information of smartphone camera specification we used in this study as: selected smartphones which support RAW (DNG) data above Android 7.0, e.g. Asus ZenFone 3 (Sony IMX298, a 16 MP camera, 4608 x 3456 pixels under Android 7.0) and/or Xiaomi Mi11 Lite 5G (Samsung ISOCELL GW3, a 64 MP camera under Android 11) (Smartphone camera info: https://zh.wikipedia.org/wiki/Exmor),(Line 77-80). In addition, with a simple magnification optical path (Figure 1B), tested smartphones can be just put on the device position and take photos as daily life shooting without further calibrations. We added the description in legends of Figure S2B as: Android Studio was used through the Java language as an integrated development environment to construct the Android App, iMet-lead. To help users easily proceed Pb sensing with pMet-lead, iMet-lead can preview the operation procedure before sensing analysis. At the stage of preview and the following data acquisition, images of YFP and CFP will be seen on the screen of  the smartphone under the interface of iMet-lead. The Ratio bottom can proceed both image acquisition (DNG and JPG format) and image ratio (YFP/CFP, only in DNG format) through smartphone camera and display in color map (the rainbow color plate is the predestined setting). Such kind of visualization can help users to easily judge whether the sample is contaminated with Pb. (Figure S2B. Line 24-30).
Response 2, 3, 6: We thank the reviewer's suggestions. On the issue of the checkerboard pattern (only appears in raw file format (DNG in 10 bit; e.g. left: Figure S3Eb), after examination through at least two different brands of android smartphone cameras on general daily life photos shown as follows (Figure S3F, by Xiaomi Mi11 Lite 5G), this DNG-only pattern still exists. It is actually in single pixel scale (left: 6 x 6 in Figure S3Eb; right and lower left: 10 x 10 in Figure 3A), but not in any color channel (left: Figure S3Ea and S3Fa, in JPG format). We think that this unique pattern originally comes from the smartphone camera itself (probably the Bayer filter which demosaicing algorithm can solve this problem for JPG color photo images). Although this pattern is unrelated to the YFP or CFP expressing cell samples taken under pMet-lead, it indeed affects the FRET ratio values during Pb sensing, since the intensity of both fluorescent images (YFP and CFP) would not be reliable at some of the pixel points (the black one shown in Figure 3A). In this view, we don’t think that background correction can repair these checkerboard pattern-induced ratio errors. As to the stripe pattern of Figure 3D (lower middle), we found it is an artifact through image handling, not the above checkerboard pattern issue. We correct this error as follows (lower right, Figure 3).
Response 4, 5: We thank the reviewer’s suggestions. In the new version of Figure 1, we added the information of the BF LED (Figure 1A).  And we also added an illustration image (Figure 1B) to show the FRET-based Pb sensing method. As to the BF and fluorescent images (Figure 1C), we provide more BF image comparison between the two image systems (pMet-lead and FRET microscope) on the left side with scale bars and let the representative fluorescent images on the right side with their own scale bar.

Figure 1. Design and layout of the portable FRET-based Pb sensing device (pMet-lead) and examples of images generated with the pMet-lead compared to conventional FRET ratio imaging. (A) Side view (upper), exploded view (middle) and entity (bottom) of the optical design of pMet-lead to demonstrate the general setup of pMet-lead: 1. the top plate holds the smartphone; 2. the middle part of the apparatus contains the plano-convex lens and is linked to the filter turret (part 4); 3. the bottom part contains the illumination sources (LED for BF, LASER Diode for FL and Light Switch to turn off (O) or on to BF (I) or FL (II)) and is linked to the sample stage (part 5); 4. the filter turret contains two emission fluorescence filters that receive YFP and CFP emission signals (changeable according to various FRET pairs); 5. the sample stage receives the biochip containing the cells expressing the Met-lead biosensor. (B) Molecular structure design (left, originally from [8], and permission has been obtained to use the same here) and emission spectrum (right) of FRET-based Met-lead. (C) The representative bright-field (BF, left) and fluorescence (YFP, middle; CFP, right) images of the Pb biosensor-expressing HEK293 cells obtained with pMet-lead (above) or under a FRET microscope (with 10 x objective; below). Scale bar, 30 μm.
Response 7: We thank the reviewer's comments. We have added three concentration points between 10 and 50 nM, i.e. 20, 30, and 40 nM as shown in the new version of Figure 4C, and the linearity (R2) was largely improved. The original data set (titration) contained totally 7 concentration points (control,1, 10, 50, 100 nM; 1 and 100 uM) is shown in Figure S4 (right). These results are very similar to that done recently (left, Biosen Bioelec 2020). Actually, it can be observed that there exist two possible linear areas: one from 10 nM to 100 nM (low Pb level) and one from 100 nM to 10 uM (high Pb level). Anyway, we focused on the low Pb level as added in the text (at the concentration range from 1 nM to 100 nM) since high concentration (above 50 nM) is sure for Pb warning.
Response 8: We thank the reviewer’s suggestions. Previously (Toxicol Sci 2012) we have tested the effects of zinc (Zn, in various concentrations from 1, 10, and  100 nM, up to 1, 10, and 100 μM) onto Met-leads in vitro, compared with lead (Pb) (left). We also did the Pb sensing with the existence of Zn (middle, 1 μM; right, 0.01, 0.1, and 1 μM). We concluded that Zn will increase the baseline of ratio values. The data in this study (Figure 5) confirmed this prediction (bottom, black bars/Pre as basal level and gray bars/Add as sensing results) and further demonstrated the influence of Zn on Pb detections both from laboratory water (Figure 5A and 5B) and from real samples (Figure 5C and 5D). Tricine was applied as a specific Zn (but not Pb) chelator to maintain the basal level of Met-lead ratio (Zn+tricine in Figure 5B; Sample+tricine in Figure 5D). The unknown real sample (Figure 5C and 5D) was later confirmed to contain both Zn and Pb (with ICP-MS validation in Figure S5B: Zn: 11 uM; Pb: 9.75 nM below the WHO regulation for Pb contamination). Thus, premixed tricine can eliminate the interference of Zn and help to detect Pb correctly and specifically. We added the data on the real sample in Figure 5C and D (Sample+tricine+Pb) to show that pMet-lead can still have ratio increase when additional Pb 10 μM was applied. The data together indicate that if there’s no tricine, the ratio increases won’t represent the correct Pb detection (false positive).

Reviewer 3 Report

Testing for Pb is indeed a serious issue for many countries.  The paper reports on a device that could apparently be made from a cell phone and with 3D printing. Sadly I have no experience with this but the authors assert that they have spent years trying to devise such a device.  The device itself is not the gold standard but I think it could be very useful since it can assess concentrations around10 pbb.  Given the alleged simplicity of the device, one wonders if plans for industrial production are in progress.  

The linearity in Fig 4c at 0.8873 was not great.  Were these measurements repeated?  There are perhaps error bars in Fig 4c but it was not clear why this low R-value pertains.  Also, the authors should test more concentrations from 5-15 at every 2ppb rather than the 4 values.

The number of runs for the samples in Fig 6 should also be stated.

Author Response

We thank the reviewer's comments. These data are indeed repeated thrice as described in Line 140-142 (All experiments were carried out three times with at least three different sets of tested samples) so are with error bars. We have added three concentration points between 10 and 50 nM, i.e. 20, 30, and 40 nM as shown in the new version of Figure 4C, and the linearity was largely improved. As to the data shown in Figure 6, it is also described in Line 140-142 (three runs).

Reviewer 4 Report

1. Many feasible Pb sensors rather than FRET such as these listed below have been reported recently. However, most of the cited literatures were self-citations. Please conduct a more complete literature review and compare the performances and commercialization feasibility in Table 3.

  • Wang, Y. Q. Pan, J. F. Wang, Y. Zhang, Y. J. Ding, A highly selective and sensitive half-salamo-based fluorescent chemosensor for sequential detection of Pb(II) ion and Cys,
  • Journal of Photochemistry and Photobiology A: Chemistry, 400, 112719, 2020.
  • Salman, H. Znad, N. Hasan, M. Hasan, Optimization of innovative composite sensor for Pb(II) detection and capturing from water samples, Microchemical Journal, 160, 105765, 2021.
  • B. Gumpu, M. Veerapandian, U. M. Krishnan, J. B. B. Rayappan, Simultaneous electrochemical detection of Cd(II), Pb(II), As(III) and Hg(II) ions using ruthenium(II)-textured graphene oxide nanocomposite, Talanta, 162, 574-582, 2017.
  • Zhu, B. Liu, H. Hou, Z. Huang, K. M. Zeinu, L. Huang, X. Yuan, D. Guo, J. Hu, J. Yang, Alkaline intercalation of Ti3C2 MXene for simultaneous electrochemical detection of Cd(II), Pb(II), Cu(II) and Hg(II), Electrochimica Acta, 248, 46-57, 2017.
  • Lu, X. Liang, J. Xu, Z. Zhao, G. Tian, Synthesis of CuZrO3 nanocomposites/graphene and their application in modified electrodes for the co-detection of trace Pb(II) and Cd(II), Sensors and Actuators B: Chemical, 273, 1146-1155, 2018.
  • Sarfo, E. Izake, A. O’Mullane, G. Ayoko, Molecular recognition and detection of Pb(II) ions in water by aminobenzo-18-crown-6 immobilised onto a nanostructured SERS substrate, Sensors and Actuators B: Chemical, 255, 1945-1952, 2018.
  • F. Sun, J. Wang, P. H. Li, M. Yang, X. J. Huang, Highly sensitive electrochemical detection of Pb(II) based on excellent adsorption and surface Ni(II)/Ni(III) cycle of porous flower-like NiO/rGO nanocomposite, Sensors and Actuators B: Chemical, 292, 136-147, 2019.

2. Please further explain the role of the human living cells in the proposed portable Pb sensing device. For practical applications, how can the users conduct the cell culture?

3. In Figure 4B, please explain why the emission ratio of the 10 nM sample is less than that of the 1 nM sample (false positive?). For the concentration ranges from 50 nM to 10 mM, the emission ratio is nonlinear (maybe linear in Log scale), how could you obtain Figure 4C from 4B? How to calculate the LOD?

4. The linearity in Line 331 and in Figure 6D are different.

5. Is Zn the only substance to interfere the Pb detection?

Wei, C. Gao, F. L. Meng, H. H. Li, L. Wang, J. H. Liu, and X. J. Huang, SnO2/reduced graphene oxide nanocomposite for the simultaneous electrochemical detection of cadmium(II), lead(II), copper(II), and mercury(II): An interesting favorable mutual interference, J. Phys. Chem. C, 116, 1, 1034–1041, 2012.

6. In Table 2, why Sample I and III with larger Y/C ratios than Sample II could not be detected by the ICP-MS? Please specify the application limitations of the proposed device.

Author Response

Response 1: We thank the reviewer's comments. We add the above 7 important papers into the new Table 3. 
Response 2: We thank the reviewer's comments. The living cells expressing the Pb biosensors on the sensing chip are indeed the central heart of the sensing device. The idea is that the users can have the cells provided by us and keep them for a short time at room temperature within 6 hours to one days will be fine after getting the sensing biochip (maybe in the future there will be some novel device that can maintain living cell culture for at least 3 to 6 days). We agree that it is somehow still a little bit inconvenient for practical application of pMet-lead. So far this is the best we can do with the sensing chip. For this, we mentioned the bacteria-expression and/or purified Met-lead will be the next generation of pMet-lead in the future at the end of the discussion part.
Response 3, 6: We thank the reviewer's comments. In Table 1 of this study, only the ratio values of 50 nM (10 ppb) reach to be significantly larger than that of control samples (4.3811 ± 0.0994 vs 2.7969 ± 0.1001). Thus we define the LOD of pMet-lead as 50 nM. In this way, any ratio values lower than 3 including those 1 and 10 nM in Figure 4B (We did the statistical comparison with these two ratio data: 1 nM and 10 nM, and found no significant difference between them) and also including the three real samples in Table 2 (even one of them was detected to have very low amounts of Pb: 0.64 ppb under ICP-MS) will be excluded (Pb undetectable) in this device. In other words, any sample containing lower than 10 ppb will not be detected by our device, and this is the true limitation of pMet-lead. The original data set (titration) contained totally 7 concentration points (control,1, 10, 50, 100 nM; 1 and 100 uM) is shown in Figure S4 (right). These results are very similar to that done recently (left, Biosen Bioelec 2020). Actually, it can be observed that there exist two possible linear areas: one from 10 nM to 100 nM (low Pb level) and one from 100 nM to 10 μM (high Pb level). We focused on the low Pb level as added in the text (at the concentration range from 1 nM to 100 nM) to present Figure 4C since high concentration (above 50 nM) is sure for Pb warning.
Response 4: We thank the reviewer's comments. We corrected this typo.
Response 5: We thank the reviewer's comments. Indeed metals other than zinc (Zn) like copper (Cu), mercury (Hg) and cadmium (Cd) might interfere with the Pb detection of Met-lead. We previously confirmed that Cu will interfere with the ratio values of Met-leads (left in Toxicol Sci 2012; right in Biosen Bioelec 2020).  From previous reports, Hg and Cd could also interfere with the ratio response of Met-lead. We think the strategy of using metal chelators similar to tricine (to Zn), such as penicillamine to Cu, DMPS to Hg, 3-((5-(trifluoromethyl)-1,3,4-thiadiazol-2-yl)amino)benzo[d]isothiazole 1,1-dioxide (AL14) to Cd will be a good solution combined inside pMet-lead in the future. So far the tested samples from Figure 6 were validated by ICP-MS to confirm without the existence of Cu, Hg, and Cd. While it will not be lucky when facing the real samples like wastewater. We will continue to carry on solving this problem in the future. And we also highlighted this issue in the new version of the manuscript as: On the issue of metal selectivity, copper (Cu), mercury (Hg) and cadmium (Cd) might interfere with the Pb detection of Met-lead in addition to Zn. It would be necessary to have a proper solution in case of applying pMet-lead to wastewater Pb detections. The strategy of using metal chelators similar to tricine (Zn), such as penicillamine to Cu [51], DMPS to Hg [52], 3-((5-(trifluoromethyl)-1,3,4-thiadiazol-2-yl)amino)benzo[d]isothiazole 1,1-dioxide (AL14) [53] to Cd will be a good solution combined inside pMet-lead in the future.

Round 2

Reviewer 2 Report

The paper is significantly improved. I recommend publication in its current form.

Author Response

We thank the reviewer's comment.

Reviewer 4 Report

1.In general, LOD is estimated using 3 sigma theory. Please use 3 sigma theory to calculate the LOD.

2. For real application, cell culture based detection approach is not so feasible. Please propose some other feasible solutions.

Author Response

Point 1: Limit of detection by 3 sigma calculation In general, LOD is estimated using 3 sigma theory. Please use 3 sigma theory to calculate the LOD.
Response 1: We thank the reviewer's suggestion. We proceeded the 3 sigma theory to calculate the LOD of pMet-lead as LOD = 3.3(Sy/S). The standard deviation of the ratio value (Sy) at 2.85 ppb (which is the lowest concentration ICP-MS can detect) was 0.0158, and the slope (S) was 0.1 obtained with one order of magnitude of calibration curve (Figure 4C). Thus the LOD = 4.74. We corrected this in Abstract as: … (Kd) 45.62 nM, and limit of detection 24 nM (0.474 μg/dL, 4.74 ppb). (Line 18) and in Results as: … , while the LOD of the new portable device was calculated as 4.74 ppb (24 nM) (Figure 4C). (Line 249).
Point 2: Biochip issue For real application, cell culture based detection approach is not so feasible. Please propose some other feasible solutions.
Response 2: We thank the reviewer's comment. On the issue of cell culture based detection approach (protein biochip and bacteria biochip as manuscript mentioned; Line 388-402), we explained more specifically about the protein-coated biochip in the new manuscript as: Finally, the biochip containing the Met-lead 1.44 M1-expressing living cells is the best one that we can reliably provide at present, but could be optimized further. To further improve portability, the biochip will be coated with purified Met-lead proteins and they will be lyophilized for longer storage, e.g. for 1 year [17].
